# Micropropagation of Apple Cultivars ‘Golden Delicious’ and ‘Royal Gala’ in Bioreactors

**DOI:** 10.3390/plants14172740

**Published:** 2025-09-02

**Authors:** Simón Miranda, Mickael Malnoy, Anxela Aldrey, María José Cernadas, Conchi Sánchez, Bruce Christie, Nieves Vidal

**Affiliations:** 1Research and Innovation Centre, Fondazione Edmund Mach (FEM), Via Mach 1, 38098 San Michele all’Adige, Italy; simondavid.mirandachavez@fmach.it (S.M.); mickael.malnoy@fmach.it (M.M.); 2Misión Biológica de Galicia, Consejo Superior de Investigaciones Científicas, Avda de Vigo s/n, 15705 Santiago de Compostela, Spain; anxela.aldrey@mbg.csic.es (A.A.); maria.jose.cernadas.cernadas@mbg.csic.es (M.J.C.); conchi@mbg.csic.es (C.S.); 3The Greenplant Company, Palmerston North 4410, New Zealand; brucechristie101@gmail.com

**Keywords:** hyperhydricity, in vitro, *Malus domestica*, multiplication, rooting, temporary immersion

## Abstract

This study aimed to investigate culture conditions for the efficient micropropagation of apple cultivars ‘Golden Delicious’ and ‘Royal Gala’ in liquid medium by temporary immersion. RITA^®^ bioreactors were used for the multiplication stage whereas RITA^®^ or Plantform™ were used for the rooting stage. Murashige and Skoog media (MS) with N^6^-benzyladenine (BA) was used for shoot multiplication and indole-3-butyric acid (IBA) for root induction. During the multiplication phase, we evaluated the mineral medium, BA concentration, immersion frequency, silver nitrate and activated charcoal supplementation and the use of physical supports to hold explants in an upright position. The results demonstrated that longer incubation periods (10 weeks) were better than shorter periods (6 weeks) for decreasing hyperhydricity and increasing the multiplication coefficient (MC). For ‘Golden Delicious’, the highest MC were obtained either with explants placed directly on the bioreactor basket and immersed six times per day for 60 s in MS with 2.2 µM BA or explants placed between rockwool cubes cultivated with 4.4 µM BA (both yielding MC of 8.9 and 5–10% hyperhydricity). These results were superior to ‘Royal Gala’, which showed a MC of 7.3 and 23% of hyperhydricity when cultivated in MS with half nitrates, 1.55 µM BA and rockwool cubes. Both varieties rooted efficiently (96–100%), and resulting plantlets were successfully acclimated. This is the first report in the micropropagation of these two commercial fruiting cultivars in temporary immersion, demonstrating the potential of this technology to enhance large-scale plant production for the apple nursery industry.

## 1. Introduction

Apple (*Malus domestica* L.) is indisputably the most cultivated fruit crop among the several edible fruits coming from the family Rosaceae, which includes pear (*Pyrus communis* L.), apricot (*Prunus armeniaca* L.), plum (*P. domestica* L.), peach (*P. persica* L.) and cherry (*P. avium* L.) [1]. It represents a good source of dietary fiber and is rich in vitamin C, folic acid, flavonoids and polyphenols, making apple consumption of nutritional value with beneficial effects in human health [2]. This pip fruit is one of the most widely cultivated crop worldwide, only preceded by watermelon (*Citrullus lanatus* (Thunberg) Matsumura & Nakai) and banana (*Musa* sp.) [3]. Global production of apple was estimated to be approximately 97 million tons in 2022–2023 with a global production value of USD 91 million [4]. In addition to health benefits, a range of other quality traits of apple fruit, including appearance, flavor, composition, texture and shelf life, are relevant to consumers when choosing specific cultivars. Among them, ‘Golden Delicious’ and ‘Royal Gala’ are two major commercially important cultivars worldwide [5]. 

The high degree of heterozygosity in the genomes of ‘Golden Delicious’ and ‘Royal Gala’ [6,7,8] represents a challenge in maintaining each cultivar that can be overcome through vegetative propagation. This approach is essential for producing genetically consistent scions and rootstocks in commercial apple cultivation. Traditionally, the main propagation method used in fruit tree nurseries to produce apple plants is grafting scions onto rootstocks, and rather less frequently may include micropropagation in semisolid medium (SSM) for plant multiplication [9]. However, tissue culture is an essential tool in the development and establishment of molecular breeding and biotechnology [9], allowing genetic improvement of apple in agronomically important traits, including resistance to biotic and abiotic stress, as well as flowering time, fruit color and modified metabolism [10]. Additionally, tissue culture protocols play an important role in the conservation of genetic resources and the production of disease-free plant material [11,12]. Micropropagation in SSM has been extensively used for many apple cultivars and rootstocks, including ‘Royal Gala’ and ‘Golden Delicious’ [9,13,14,15,16]. Similarly, new breeding techniques, such as gene editing by CRISPR/Cas9 and the production of marker-free transgenic plants are well established in these cultivars [17,18], which also rely on micropropagation in SSM for multiplication of shoots used for *Agrobacterium*-mediated transformation. At present, the in vitro propagation conditions and delivery of editing components into apple cultivars require further optimization to improve the applicability of new breeding techniques.

Propagation in bioreactors is emerging as a promising approach to improve plant proliferation and physiological condition [19], and to assist novel technologies including nanoparticle delivery, sanitation, genetic transformation and general propagation improvement [20,21,22]. Bioreactor use for micropropagation has proven effective in many fruit trees, including cherry and *Prunus* rootstock [23,24], plum [25] and chestnut (*Castanea sativa* Mill.) [26]. Although the potential of this technique for apple propagation has been recognized for more than a decade [27], to the best of our knowledge, the studies for apple micropropagation in bioreactors have been limited to M9, M16, J9 and M26 rootstocks [28,29,30,31], varieties such as *Malus* × *domestica* ‘Holsteiner Cox’ [32], Carrandona, Florina and Rozona [24] and *Malus sylvestris* [33], without reports of protocols based on liquid media for ‘Golden Delicious’ and ‘Royal Gala’. 

The present study represents a collaboration of the researchers of FEM (Italy) and MBG (Spain) in the frame of the COST Action CA21157, “European Network for Innovative Woody Plant Cloning”. We exchanged plant material and used the previous experience of our research groups in applying biotechnological tools to improve apple micropropagation, by assessing and optimizing the conditions to culture commercially important apple cultivars ‘Golden Delicious’ and ‘Royal Gala’ in bioreactors. The main aim of this study was to investigate the culture conditions that can improve the growth of these cultivars in bioreactors and allow future applications, including exogenous treatments with bioactive compounds.

We used different systems and bioreactors for the multiplication and rooting stages of micropropagation. During the multiplication phase, a temporary immersion system (TIS) was used. We employed RITA^®^ bioreactors, which were effective in propagating other apple genotypes [29]. RITA^®^ bioreactors can be a good choice to initiate studies with new species or genotypes in liquid medium. They use fewer explants than other commercial bioreactors like Plantform™, ElecTIS or SETIS™ [21,33,34], allowing a higher number of treatments to be compared at the same time [35]. In the multiplication stage we evaluated the effect of several factors, including mineral medium, cytokinin concentration, immersion frequency, silver nitrate and activated charcoal supplementation, the use of physical supports to hold explants in an upright position and the culture duration. We used different nutrient solutions for ‘Golden Delicious’ and ‘Royal Gala’, corresponding to the formulations adapted to each cultivar and routinely used in FEM laboratories for their micropropagation in semisolid medium. Genotype specificity for nutrients and plant growth regulators requirements are well referenced in apple [9,27,36]. In this study, in ‘Royal Gala’ we investigated the use of full-strength Murashige and Skoog (MS) medium [37] or MS with nitrogen reduced to a half (MS ½ N), whereas regarding plant growth regulators, we evaluated the effect of varying the cytokinin concentration using N^6^-benzyladenine (BA) in both cultivars. 

Adventitious rooting was induced using indole-3-butyric acid (IBA). We used various culture systems and bioreactors to take advantage of the distinct characteristics of the containers to find suitable alternatives for two objectives: (i) developing protocols for large-scale plant production and (ii) developing methods to monitor the uptake of bioactive compounds in vitro for future experiments on plant physiology. Shoots were rooted in jars in semisolid medium or in liquid medium in bioreactors operated either by TIS or by continuous immersion system (CIS). RITA^®^ bioreactors were operated always by TIS, whereas Plantform™ were operated by TIS—as recommended by the manufacturer [38]—or by CIS [25]. In the latter case the internal elements of the bioreactors were eliminated but the container received aerations at fixed intervals [39]. The basal sections of shoots were placed on the bottom of the container in contact with the liquid medium. For rooting in CIS and TIS, supports were used to keep shoots in upright position. 

Our results demonstrated that ‘Golden Delicious’ and ‘Royal Gala’ can be successfully propagated in bioreactors. ‘Golden Delicious’ showed greater proliferation rates and showed lower levels of hyperhydricity compared to ‘Royal Gala’. Both varieties rooted efficiently, and resulting plantlets were successfully acclimated. 

## 2. Results

### 2.1. First Experiments with Golden Delicious and Royal Gala in Bioreactors: Effect of Support

Preliminary trials with apple in liquid medium showed varying degrees of hyperhydricity depending on the cultivar (Figure 1). 

To mitigate this problem, we carried out an experiment to evaluate the impact of physical supports on the growth of both cultivars. Apical shoots of ‘Golden Delicious’ and ‘Royal Gala’ previously cultured in jars were placed either directly onto RITA^®^ baskets (without support), between 1 cm^3^ rockwool cubes, or inserted into plastic slots in a custom-made holder (Figure 2). 

For each cultivar, we used the same medium composition as previously used in jars, but omitting agar. Table 1 presents the results obtained after 6 weeks of immersing apical sections of both cultivars for 90 s six times per day. 

Results from Table 1 indicate that ‘Golden Delicious’ can grow under all tested conditions, with the use of a support significantly improving the multiplication coefficient (MC). Both rockwool cubes and slots effectively reduced HH and increased MC, with no statistically significant differences between them. As inserting explants into slots is more time-consuming and technically demanding than placing them between the rockwool cubes, the latter option was selected for the next experiments.

In contrast with ‘Golden Delicious’, ‘Royal Gala’ produced mainly hyperhydric shoots in all the treatments. To reduce HH and enable multiplication in this cultivar, we conducted an experiment using rockwool cubes and shortened the immersion duration to 60 s. The investigated factors were frequency of immersion (six, three or two immersions per day), BA concentration (3.10 µM or 1.55 µM) and the nitrogen content of the mineral medium (full-strength MS or MS with nitrogen reduced to a half (MS ½ N)). Results are shown in Table 2.

None of the treatments with full-strength MS showed positive responses, as HH was over 80% and MC under 1.0. However, the combination of MS ½ N with 1.55 µM BA allowed it to reduce HH percentages to about 40%, and to obtain MC values higher than 3 (with six immersions per day). Less than six immersions further reduced the frequency of hyperhydric shoots (20 and 26%), but this was accompanied with a reduction in the total shoot number and a decrease in MC (2 and 2.5 instead of 3.5). For this reason, the combination of MS ½ N with 1.55 µM BA and six immersions per day was used in the next experiments with ‘Royal Gala’. 

### 2.2. Optimization of Multiplication of ‘Golden Delicious’ in Bioreactors

#### 2.2.1. Effect of Cytokinin Concentration, Subculture Duration and Support

In the first experiments with liquid medium, we observed the occurrence of HH, which could be mitigated by using supports maintaining explants in an upright position (Table 1). We then examined the combined effect of reducing cytokinin supplementation and extending the subculture duration from 6 to 10 weeks, together with the use of physical supports. 

The three factors investigated (BA concentration, subculture duration and support) had significant effects on shoot multiplication and quality (Figure 3). 

Extending the subculture duration from 6 to 10 weeks improved multiplication coefficient and number of normal and rootable shoots (*p* < 0.001 for MC and normal shoot number and *p* = 0.006 for rootable shoot number). Results were influenced by the interaction between BA supplementation and the use of cubes after 10 weeks of culture (Figure 3b,d,f) (*p* < 0.001 for MC, normal shoot number and rootable shoot number). Hyperhydricity was markedly reduced in treatments with rockwool cubes, regardless of cytokinin level or duration, with the lowest values observed at 2.2 µM BA after 10 weeks (Figure 3c). However, when shoots were grown with 2.2 µM BA, rockwool cubes had a negative effect on the number of total shoots (Figure 3a), which in turn reduced the number of normal shoots, MC and the number of shoots useable for rooting (Figure 3b,d,f). Interestingly, the best responses—MC of 8.9 and 8.8 and number of rootable shoots of 2.8 and 3.2—were very similar and corresponded to inversely related treatments: 2.2 µM BA without cubes and 4.4 µM BA with cubes after 10 weeks of culture (Figure 3d,f). These findings indicate that the control of HH—either by lowering the cytokinin supplementation or by using supports to maintain the shoots in upright position—is a key factor for the successful propagation of ‘Golden Delicious’ in bioreactors. The next subsection explores other approaches to HH control.

#### 2.2.2. Strategies to Reduce Hyperhydricity in ‘Golden Delicious’: Using Silver Nitrate and Activated Charcoal

Table 3 shows the effect of including silver nitrate (SN) or activated charcoal (AC) in the liquid medium in RITA^®^ bioreactors. Shoots were treated with 4.4 µM BA and results recorded after 10 weeks. In the treatment conditions used in the study, both SN and AC were effective in reducing HH, although in the case of SN it only occurred for shoots cultured without support. Among shoots cultured without support, SN produced more normal shoots and higher MC than the controls. However, SN treatment significantly reduced the number of total shoots relative to controls (3.1 and 3.6 instead of 5.5 and 5.9 for explants cultured without and with cubes, respectively). As a consequence, for the number of normal shoots, MC and the number of rootable shoots, the values obtained with SN (with or without cubes) were always lower than those shown by controls cultured with cubes.

Similarly, AC eliminated HH completely, but at the cost of markedly reducing the number and length of shoots, with a highly significant reduction in MC and the number of rootable shoots. Interestingly, shoots treated with AC developed spontaneous roots in the multiplication medium (Figure 4), reaching percentages of 54% for explants cultured with cubes. Another feature of SN and AC treatments was a significant increase in the leaf size (Table 3, Figure 4), which was more marked in the case of AC. 

### 2.3. Optimization of Multiplication of ‘Royal Gala’ in Bioreactors

#### 2.3.1. Effect of Support and Subculture Duration

Figure 5 shows the effect of culturing apical sections of ‘Royal Gala’ in RITA^®^ bioreactors with MS ½ N and 1.55 µM BA. Shoots were placed directly on the bioreactor baskets or between rockwool cubes, and data were recorded after 6 and 10 weeks. Results were similar to the experiment with ‘Golden Delicious’ (Figure 3): extending the subculture duration increased the multiplication and overall performance of this cultivar. With the exception of shoot length and rootable shoot number, where the treatment effects were not statistically significant, the rest of variables showed a significant improvement with the longest culture period. Cubes use had a marked positive effect on most of the variables (Figure 5d–f). At 6 weeks, explants cultivated between cubes showed twice the MC as those grown without support (4.8 versus 2.4), and at 10 weeks almost 70% more (7.3 versus 4.3). With cubes, shoots grew longer (Figure 5e) and rootable shoots number increased either 3 or 4 times depending on the culture duration (Figure 5f). Interestingly, cubes did not affect HH percentages (Figure 5c). The proportion of HH shoots ranged from 23 to 36%, which depended only on the subculture duration. However, even the lowest value of HH observed in this experiment (23%) represents a considerable loss of propagation material, reducing the potential of TIS for mass production of ‘Royal Gala’. To decrease the HH percentages we performed the experiment described in the next subsection. 

#### 2.3.2. Strategies to Reduce Hyperhydricity in ‘Royal Gala’: Using Silver Nitrate and Activated Charcoal

Table 4 shows the effect of including silver nitrate (SN) or activated charcoal (AC) combined or separately on the performance of ‘Royal Gala’ shoots cultured for 10 weeks in bioreactors. Explants were placed directly on the bioreactor basket or between cubes and were immersed six times per day in MS ½ N supplemented with 1.55 µM BA. Silver nitrate and activated charcoal affected the development of ‘Royal Gala’ shoots, but none of the treatments surpassed the values shown by the control grown using cubes (Table 4). In fact, SN increased the proportion of HH shoots, almost triplicating the control values when explants were cultured without cubes (62 versus 23). As observed previously in ‘Golden Delicious’, AC reduced HH when applied independently to explants placed between cubes (4% versus 24% of the control) but the number of shoots was also significantly reduced, causing a drastic drop in MC values (2 versus 7). 

With ‘Royal Gala’ we also investigated the effect of combining the two compounds, but the outcomes were not satisfactory and did not improve plant performance, with SN increasing the percentage of HH shoots (Table 4, Figure 6a). The only variables that showed a positive response with the use of SN and AC were the leaf size, which showed significant increments especially in the treatments including AC, and the number of shoots that rooted spontaneously in the multiplication conditions. Most of the explants treated with AC developed primary roots with secondary growth that made them useful for direct acclimation (Figure 6b). 

### 2.4. Rooting and Acclimation of Golden Delicious and Royal Gala

Vigorous shoots of the two cultivars were used to evaluate rooting in liquid medium in bioreactors. Shoots rooted in semisolid medium in jars were used as controls. Rooting treatments consisted of the combination of bioreactors (RITA^®^ and Plantform™), culture systems (TIS and CIS) and supports to maintain the shoots in an upright position during the process (plastic slots, and cubes of 1 or 2 cm^3^ size, where shoots could be inserted (2 cm^3^) or placed between them (1 cm^3^)). Results from treating shoots of the 2 cultivars with 4.9 µM IBA for 5 weeks are shown in Figure 7. 

‘Golden Delicious’ and ‘Royal Gala’ showed high percentages of rooting in all the treatments in liquid medium (Figure 7), with values between 70 and 100% in ‘Golden Delicious’ (*p* = 0.030) and between 75 and 96% in ‘Royal Gala’ (*p* = 0.275) that were at least as good as those observed in jars. The best conditions for rooting were Plantform™ operated by TIS and cubes of 1 cm^3^ for ‘Golden Delicious’ and Plantform™ operated by CIS and cubes of 2 cm^3^ for ‘Royal Gala’. The morphology of the root system can be observed in Figure 8. Rooted shoots were successfully transferred to soil-less media for acclimation in a phytotron for 4 weeks followed for 3 months in the greenhouse.

## 3. Discussion

In this study, we investigated the factors influencing the micropropagation of two apple cultivars in bioreactors, ‘Golden Delicious’ and ‘Royal Gala’. The experiments allowed us to propose slightly different propagation protocols for each of them, reaching multiplication coefficients of 8.9 and 7.3, and rooting percentages of 100% and 96% for ‘Golden Delicious’ and ‘Royal Gala’, respectively. 

The multiplication step was carried out by temporary immersion in RITA^®^ bioreactors operated as recommended by the manufacturers. For the rooting step we compared RITA^®^ and Plantform™ bioreactors, the operation mode (TIS or CIS), as well as using different supports. 

We detected differences in the multiplication requirements for ‘Golden Delicious’ and ‘Royal Gala’ in bioreactors, even though they are closely related genetically [40]. Genotypic differences in growth and developmental responses have been reported extensively for apple and mineral media composition, type and concentration of growth regulators, vitamins and other organic compounds that require adjustment for different genotypes [27,36,41]. In this study, ‘Golden Delicious’ responded positively to liquid culture from the first experiments, while ‘Royal Gala’ presented the challenge of managing hyperhydricity. This disorder represents one of the most common problems in plant micropropagation, limiting the success of culturing plants in semisolid medium and in bioreactors. Hyperhydric shoots are characterized by anatomical and morphological abnormalities, like translucent and brittle stems and leaves, as well as physiological and metabolic dysfunctions that make them unsuitable for multiplication and rooting [42,43]. Apples often develop hyperhydricity in semisolid medium [13,44,45] and have been used as a model species for recent developments in automatic phenotyping of this disorder [46], which may prove useful for future management of this condition. 

Hyperhydricity is also one of the most important obstacles affecting success on micropropagation of apple in bioreactors [27,28,29,47]. Frequently, the degree of susceptibility is genotype-dependent, as has been reported for pear [34] and chestnut [48]. Although the causes and mechanisms of HH are not completely understood [49], there is a scientific consensus on the involvement of factors, such as water availability, cytokinin, ammonium and ethylene, that frequently are interrelated [43]. A relationship between high concentrations of ammonium in culture media and hyperhydricity has been reported for plants grown in semisolid medium [50,51,52]. It has been claimed that the negative effects of high ammonium concentrations rely on its rapid absorption from the medium, that alters the carbohydrate metabolism causing a deficiency of lignin and cellulose, which in turn increases water uptake [53]. In addition, ammonium increases ethylene production inside the vessels [54,55], which increases HH [56]. It is speculated these effects would probably be more dramatic in bioreactors, as the liquid medium increases nutrient uptake [57] and ammonium would be absorbed more easily than in semisolid medium. In this study, both cultivars developed hyperhydric shoots, but in ‘Golden Delicious’ it could be effectively controlled with the use of a support, as reported for chestnut [48]. The rockwool cubes maintained the apple shoots in an upright position, while also keeping the basal section of the shoots in contact with the medium, providing the explants with a constant supply of nutrients and, at the same time, allowing the gas exchange between the rest of the shoot and the surrounding environment, which is crucial for the normal functioning of higher plant tissues [58]. The higher expression of hyperhydricity in bioreactors could be related to the higher availability of plant growth regulators in liquid medium, especially cytokinins, that can contribute to increased ethylene biosynthesis [59,60]. This was demonstrated in kiwifruit (*Actinidia deliciosa* (Chevalier) Liang & Ferguson), where the explants cultured in liquid medium accumulated more BA in the meristematic tissues than shoots cultured in SS medium [61]. Decreasing the concentration of media cytokinin reduced hyperhydricity in the two apple cultivars in this study, as reported for chestnut, teak (*Tectona grandis* L.) and *Dianthus chinensis* L. [48,62,63]. This was especially critical for ‘Royal Gala’, which could not be propagated in liquid medium with the same BA concentration as in jars (Table 2).

We noticed that extending the period of culture from 6 to 10 weeks was beneficial as it significantly reduced HH and increased MC in both cultivars. It appears that some shoots with mild HH symptoms recovered when given the opportunity of the extended culture period. Further research is required to investigate the cause and effect of this result, as we think it might be related to gas exchange or gas composition in the culture vessel or to a decrease in certain nutrients in the liquid medium. Additional observation and investigation of the interaction between support and BA observed in ‘Golden Delicious’ after 10 weeks of culture is required. In this cultivar, multiplication coefficients and number of rootable shoots when cultured with 2.2 µM BA without support were similar to MC and rootable shoots obtained when BA was doubled and shoots were placed between cubes. However, the use of the support in explants cultured with the lowest BA concentration decreased hyperhydricity but did not increase the MC, probably because the number of total shoots in medium with 2.2 µM BA was low and HH was not so significant in this treatment. In contrast, in ‘Royal Gala’ the use of cubes did not have an impact on HH but was effective in increasing the proliferation and the number of rootable shoots. 

The beneficial effect of rockwool cubes in bioreactors has been reported for chestnut, alder (*Alnus glutinosa* (L.) Gaertn.) and pedunculate oak (*Quercus robur* L.) [48,64,65]. In those studies, the advantage of using rockwool as a support was attributed to avoiding the complete immersion of the explants in the liquid medium, but with apple, the positive effect of the cubes could be more complex, involving both the physical and chemical environment acting on the basal section of the shoots and not only their effect on maintaining the apical section above the liquid phase. The rockwool could be holding nutrients in moisture-filled pores that would be available to the shoots during the intervals between immersion and also adsorbing compounds that interfere with plant development. Furthermore, gaseous environment in the porous space in the rockwool may be positively impacting growth and development. 

To be commercially applicable, both cultivars should ideally exhibit reduced HH while maintaining a high multiplication rate, without reliance on support materials such as rockwool cubes, which despite their positive effects, increase the time of transferring the explants into the bioreactors and the risk of contamination. In this study, besides the reduction in the cytokinin concentration and the avoidance of total explant immersion commented above, we applied other strategies found in the literature that impact hyperhydricity development: (i) the reduction in immersion frequency, (ii) use of lower concentrations of ammonium, (iii) supplementation with silver nitrate and (iv) activated charcoal. 

The reduction in the immersion frequency showed good results with pioneer studies on pear [34], chestnut [48] and oak [66]. In this study, it was only applied to ‘Royal Gala’ as it presented a major challenge when cultured in bioreactors. However, the reduction in HH was very small—probably because the structure of RITA^®^ bioreactors links immersions with aerations, and a decrease in immersion means a decrease in aeration—and was concomitantly accompanied by a decrease in proliferation. In contrast, the reduction in ammonium content had a positive response in reducing HH and increasing multiplication. These experiments reduced the ammonium media component by using MS with half nitrogen (affecting NH_4_NO_3_ and KNO_3_); further experiments should be conducted to separate the effect of ammonium from the other ions (NO_3_^−^, K^+^).

The effect of silver nitrate and activated charcoal on apple was different from what was expected. SN effectively reduced HH in several species [67,68,69] as it can regulate the activity of ethylene [63,70,71]. In ‘Golden Delicious’ SN reduced HH in the explants cultured without cubes, but MC was not improved. In contrast, when SN was applied to explants cultured with cubes HH was unchanged but the number of shoots and MC values were almost halved. As ethylene in appropriate physiological concentration is beneficial for bud formation [72,73,74], the results with SN obtained in our study could suggest that the SN concentration used was excessive and may have removed part of the ethylene involved in normal shoot development. In *Dianthus chinensis*, SN concentrations similar to that used here also decreased shoot number, whereas lower concentrations enabled successful multiplication [63]. In ‘Royal Gala’ SN increased HH when applied alone, and did not have a significant effect on HH when applied along with AC. In both cases, total and normal shoots, as well as MC, were significantly reduced. Undesirable effects of silver nitrate have been related to its off-target effects, including altered auxin transport [75], but in our case could also be related to the use of an inadequate concentration. In apple, further experiments should be conducted to shed more light on how silver affects growth and development in vitro.

Activated charcoal has been used in the control of HH in vitro [76,77,78]. Although the mechanisms are not clear, it has been suggested the plant response could be related to a reduced uptake of ammonium from the medium, as demonstrated with *Lagerstroemia indica* (L.) Pers. cultivated hydroponically [79]. Alternatively, other studies suggest that the effect of AC on the mitigation of HH could be related to its capacity to adsorb ethylene [80], although this effect appears to be strongly dependent on culture conditions, such as container volume and shape, medium volume and the surface exposed to the inner atmosphere [80]. In ‘Golden Delicious’ AC eliminated HH but at the cost of decreasing the number of shoots. In ‘Royal Gala’ AC significantly reduced shoot number and MC, but its effect on HH was restricted to the explants cultured with rockwool cubes. AC and SN increased leaf size in the two cultivars, and AC had a positive effect on the formation of adventitious roots during the multiplication step, this being more evident in ‘Royal Gala’. Although the beneficial effect of AC on rooting in this cultivar has been previously reported [81], that study evaluated the effect of AC when supplemented in rooting expression medium and not in the multiplication step as in this study. Our results might suggest that AC treatment could be reducing the availability of BA and other nutritional factors [76,77]. This was indicated by the concomitant rooting response and leaf enlargement which may also have contributed to some mixo- or autotrophy. SN and AC may have beneficial effects on apple propagation, indicating the need for further research to evaluate the effect of applying them across a broader concentration range than that used in the present study. 

Apple shoots produced by temporary immersion were successfully rooted in bioreactors, without marked differences between the two cultivars. The positive effect of liquid medium for in vitro rooting of apple has been mentioned [41,82,83,84,85] but in these reports bioreactors were not used. In our study we compared different strategies for rooting. All of them could provide different insights or advantages depending on the intended use of the rooted shoots; treatments with semisolid medium allows visualization of the moment when and how roots emerge without disturbing the shoots, using RITA^®^ with plastic slots enables a rapid and easy change in medium for other biotechnological treatments, as well as an increase in uptake of bioactive substances due to the forced ventilation. This later feature is shared by the Plantform™ bioreactors when operated in the TIS mode, with the additional benefit that they can hold a larger number of explants. Finally, shoots inserted in 2 cm^3^ rockwool cubes in Plantform™ bioreactors when operated by TIS or CIS are easier to handle without causing root damage. This is important, if the purpose is to transfer them to soil-less media for acclimation and large-scale plant production. The use of fibrous or porous support materials for rooting has been recommended as a simple and cost-effective means of micropropagation [86,87] and proved beneficial for other woody plants such as American chestnut (*Castanea dentata* (Marshall)) [88,89] and European chestnut [26,39], peach [90] and plum [25]. In our case we obtained high rooting rates with all the treatments and the plantlets were successfully acclimated.

Future research will evaluate different cytokinins as well as the roles of silver, activated charcoal and nitrogen source on apple propagation. The use of Plantform™ bioreactors, aeration regime and CO_2_ supplementation will also be investigated during the multiplication stage of ‘Golden Delicious’ and ‘Royal Gala’. Overall, this study identified key factors and conditions for optimizing of ‘Golden Delicious’ and ‘Royal Gala’. Our results can be readily applied in the fruit micropropagation industry. In addition, further research will allow us to test photoautotrophic propagation—independent of exogenous sugar supplementation—which could further enhance production by decreasing cost and residue. The use of bioreactors and forced aeration to grow apple could be implemented to evaluate the delivery and effect of other bioactive molecules, thus providing a valuable strategy to assess its potential application in genetic improvement, functional studies and sanitation technologies to these cultivars.

## 4. Materials and Methods

### 4.1. Plant Material

Initial explants for bioreactor experiments were shoots established in vitro from ‘Golden Delicious’ and ‘Royal Gala’ mother stock planted in 2011 and maintained in the field *Malus* germplasm collection of Fondazione Edmund Mach, in Trento, Italy (GPS coordinates 46.181848, 11.119849). Each genotype comprises three individuals per accession (PI 392303 and PI 590184 for cv. ‘Royal Gala’ and ‘Golden Delicious’, respectively). Initiation of in vitro cultures of both genotypes was carried out from 5 to 10 cm long young developing shoots surface-sterilized by immersion in 70% ethanol for 30 s followed by 0.1% sodium hypochlorite for 15 min and three rinses with sterile distilled water. Both genotypes were maintained by regular 6-week subcultures in SSM for three years before the initiation of the experiments of the present study. SSM was based on Murashige and Skoog medium including vitamins (MS) [37] (Duchefa Biochemie, Haarlem, the Netherlands), supplemented with 3% sucrose and solidified with 0.7% Plant Propagation Agar (Condalab, Madrid, Spain). The ‘Golden Delicious’ medium was supplemented with 60 mg/L myo inositol, 4.4 µM BA, 1.48 µM IBA and 0.58 µM Gibberellic acid (Sigma Aldrich^®^, Merck KGaA, Darmstadt, Germany). For ‘Royal Gala’ MS medium was supplemented with 3.11 µM BA and 289 µM Fe-EDDHA (Duchefa Biochemie, Haarlem, the Netherlands). The medium pH was adjusted to 5.6–5.7 prior to autoclaving at 120 °C for 20 min.

### 4.2. Growth Conditions in Bioreactors

Micropropagation experiments used for temporary immersion (TIS) and continuous immersion (CIS) were conducted in commercial RITA^®^ [91] (Vitropic, Saint Mathieu de Tréviers, France) and Plantform™ bioreactors [38] (Plant Form AB, Hjärup, Sweden). These containers were prepared, sterilized and operated as described in detail for chestnut [92]. Briefly, after assembling the bioreactors, the inlet and outlet holes were protected for sterilization. Filters were also protected and autoclaved separately. The bioreactor lids were partially closed to prevent deformation during autoclaving, and were protected with aluminum foil. When supports were used, they were previously cleaned or moistened, autoclaved, dried and kept in a clean container before being placed in the bioreactors for a second sterilization inside them. As excessive pressure can damage the bioreactors, we autoclaved bioreactors and filters at 115 °C for 20 min. In the case of RITA^®^, 20–50 mL of water was added for autoclaving and removed afterwards to help sterilization and the identification of possible cracks, whereas the medium was autoclaved separately at 120 °C for 20 min. When using Plantform™ the medium was added to the bioreactor before autoclaving. The filters were dried in an oven (50 °C) and cooled before use.

The initial explants used for TIS consisted of 15–20 mm apical sections with 2–3 leaves that were obtained from explants grown on semisolid medium for 6 weeks. Explants were cultured with 150 mL or 500 mL of medium (RITA^®^ or Plantform™, respectively). Cultures were maintained in a 16 h photoperiod provided by LED lamps with a photosynthetic photon flux (PPF) density of 50–60 µmol m^−2^ s^−1^ at 25 °C (light) and 20 °C (dark). Explants were cultured for 6 to 10 weeks, then data corresponding to shoot quality and multiplication were recorded and shoots were collected for multiplication or rooting experiments. 

Experimental factors included physical support, immersion frequency and medium composition, as well as the effect of silver nitrate (SN) and activated charcoal (AC), as detailed below.

#### 4.2.1. Physical Support

Explants were either placed directly on the bioreactor net or maintained in an upright position by placing them between 1 cm^3^ rockwool cubes (Grodan, Roermond, the Netherlands) or inserted into plastic slots in custom-made holders. In some cases, RITA^®^ baskets were partitioned with aluminum foil to accommodate different support conditions within the same vessel as described for oak [65]. In TIS and CIS rooting experiments, custom-made slots as well as rockwool cubes of 1 and 2 cm^3^ (Grodan, Roermond, The Netherlands) were evaluated. Shoots were inserted in the 2 cm^3^ cubes or placed between the 1 cm^3^ cubes as in the multiplication stage.

#### 4.2.2. Immersion Frequency

In initial TIS multiplication experiments in RITA^®^ bioreactors, apical sections were immersed for 90 s 6 times per day. In further optimization experiments, shoots were immersed for 60 s in variable frequencies ranging from 2 to 6 times per day. In rooting experiments performed in RITA^®^ bioreactors, apical sections were immersed for 60 s 6 times per day. In rooting experiments performed in Plantform™ operated by TIS the immersion regime was 6 immersions of 60 s plus 9 aerations of 60 s, whereas for CIS we applied 15 aerations of 60 s to ensure that both systems had the same air supply.

#### 4.2.3. Media Composition

Explants for TIS multiplication experiments were obtained from shoots growing on shoot propagation semisolid medium and were cultured in TIS with the same formulation but omitting agar. During optimization steps aimed to reduce hyperhydricity, full-strength MS medium or half-strength nitrates (MS ½ N) were used with variable concentration of BA (1.55–4.4 µM). In some experiments with ‘Royal Gala’ and ‘Golden Delicious’ genotypes, the medium was supplemented with 1 g/L of activated charcoal (AC) (Sigma Aldrich^®^, Merck KGaA, Darmstadt, Germany) or 30 µM AgNO_3_ (SN) (Duchefa Biochemie, Haarlem, The Netherlands). AC was added before autoclaving, while SN was filtered through a 0.22 µm filter and added after autoclaving.

#### 4.2.4. Root Induction

For rooting TIS and CIS experiments, shoots longer than 25 mm with 3–4 leaves and sub-cultured in multiplication experiments for 6 or 10 weeks were transferred to half-strength macronutrients MS medium (MS ½) supplemented with 4.9 µM indole 3-butyric acid (IBA) (Sigma Aldrich^®^, Merck KGaA, Darmstadt, Germany) and 3% sucrose. RITA^®^ with plastic slots, as well as Plantform™ bioreactors were used in TIS regime as described for multiplication steps. Plantform™ were also used by continuous immersion (CIS) as described for rooting of plum [25], in both cases with 1 or 2 cm^3^ rockwool cubes (Grodan, Roermond, The Netherlands).

### 4.3. Rooting Assessment and Acclimation

After 6 weeks of culture in rooting-inducing medium, rooted shoots were transferred to plug trays filled with a peat/perlite (3:1). Plantlets were acclimated for 4 weeks in a controlled environmental chamber (Fitotron SGC066, Sanyo Gallencamp PLC, Leicestershire, UK) with a photoperiod of 16 h light/8 h dark, a PPF density of 240–250 µmol m^−2^ s^−1^, a temperature of 25 °C (light) and 20 °C (dark) and a relative humidity of 85% and then transferred to the greenhouse.

### 4.4. Data Recording and Statistical Analysis

For data recording, the parameters analyzed in multiplication experiments were as follows: (a) the number of shoots longer than 15 mm per explant, differentiating between normal and abnormal hyperhydric shoots; (b) the percentage of the total shoots that showed hyperhydric symptoms; (c) the length of the longest normal shoot per explant; (d) multiplication coefficient, defined as the number of segments useful for multiplication obtained per initial explant; (e) the number of rootable shoots obtained per initial explant, defined as non-hyperhydric shoots longer than 20 mm and with an active apex; (f) the length and the width of the longest normal leaf per explant. For rooting experiments, the parameters recorded were (g) the percentage of rooted shoots, (h) number of roots, (i) length of the longest root and (j) shoot viability.

Data correspond to 24 explants per treatment in shoot multiplication experiments (at least two RITA^®^ vessels per repetition and two repetitions per experiment) and to 16 shoots per genotype in rooting experiments. The data were analyzed by Levene’s test (to test the homogeneity of variance) and the Shapiro–Wilk test to test normality in data distribution. The data were then subjected to one or two-way analysis of variance (ANOVA), followed by comparison of group means (Tukey-b test). When an interaction between two factors was indicated by the two-way ANOVA, Bonferroni’s adjustment was applied to detect simple main effects. When the conditions for ANOVA could not be met, data were analyzed by the Kruskal–Wallis non-parametric test. Statistical analyses were performed using SPSS 29.0 (IBM Corp., Armonk, NY, USA).

## 5. Conclusions

In this study, we demonstrated the feasibility of micropropagating ‘Golden Delicious’ and ‘Royal Gala’ in liquid medium in RITA^®^ bioreactors. The two varieties reacted differently to liquid medium: ‘Golden Delicious’ produced vigorous shoots in most conditions whereas ‘Royal Gala’ showed more hyperhydricity. The proposed protocols for multiplication consist of immersing apical shoots for 60 s six times per day for 10 weeks. The best medium for ‘Golden Delicious’ was MS with vitamins supplemented with 3% sucrose, 60 mg/L myo inositol, 1.48 µM IBA, 0.58 µM Gibberellic acid and either 2.2 µM BA (without rockwool cubes) or 4.4 µM BA (with cubes). For ‘Royal Gala’ the best medium was MS ½ N with vitamins supplemented with 3% sucrose, 1.55 µM BA and 289 µM Fe-EDDHA (with cubes). For rooting, MS ½ macronutrients with vitamins supplemented with 4.9 µM IBA in a range of systems and substrates produced 96–100% rooted shoots in both cultivars. To the best of our knowledge, our results represent the first study reporting key micropropagation factors required to propagate these two commercial fruiting cultivars in temporary immersion. Bioreactors produced successful results for multiplication, rooting and acclimation. Our results will be useful for large-scale plant production and for the assessment of potential applications in genetic improvement and functional studies in these apple cultivars.

## Figures and Tables

**Figure 1 plants-14-02740-f001:**
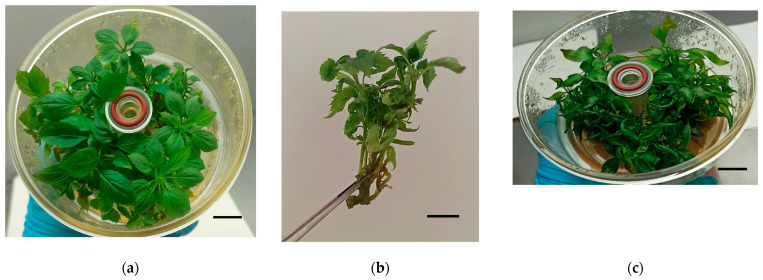
Initial experiments of apple in bioreactors. (**a**,**b**) Normal (**a**) and partially hyperhydric (**b**) shoots of ‘Golden Delicious’ after 6 weeks of culture. (**c**) Hyperhydric shoots of ‘Royal Gala’ after 6 weeks of culture. Bars = 1 cm.

**Figure 2 plants-14-02740-f002:**
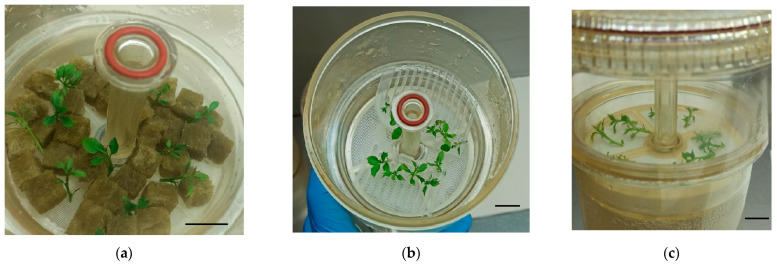
Initial explants of apple in bioreactors with or without supports in the moment of inoculation in the bioreactors. (**a**) Apical sections of ‘Royal Gala’ placed between 1 cm^3^ rockwool cubes. (**b**) Apical sections of ‘Golden Delicious’ inserted between plastic slots. (**c**) Apical sections of ‘Royal Gala’ without support. Bars = 1 cm.

**Figure 3 plants-14-02740-f003:**
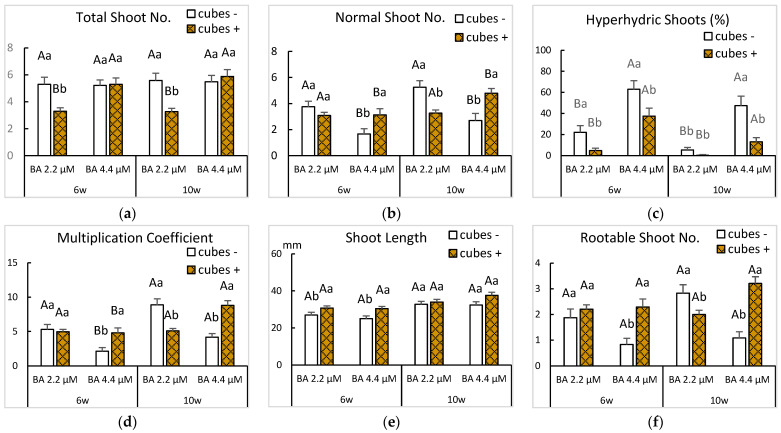
Effect of cytokinin concentration and subculture duration on the growth parameters of ‘Golden Delicious’ shoots cultured in bioreactors with or without rockwool cubes. Shoots were cultured in MS with BA 2.2 or 4.4 µM and immersed 6 times per day for 60 s. Data were recorded 6 and 10 weeks after starting the experiment. (**a**) Number of total shoots per explant. (**b**) Number of normal shoots per explant. (**c**) Percentage of hyperhydric shoots per explant. (**d**) Multiplication coefficient. (**e**) Length of the longest shoot per explant. (**f**) Number of rootable shoots per explant. Means ± standard error were calculated from 3 replicates, each with 8 shoots per treatment (*n* = 24). For each subculture duration, different uppercase letters indicate significant differences between BA supplementation and different lowercase letters indicate significant differences regarding the use of a support (*p* ≤ 0.05). A significant interaction between factors in sections (**a**–**d,f**), required a Bonferroni adjustment to detect simple main effects between means.

**Figure 4 plants-14-02740-f004:**
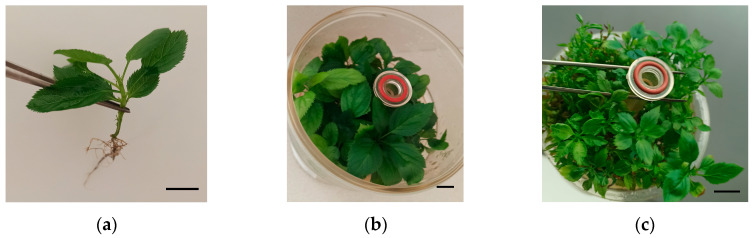
Effect of activated charcoal in shoot quality of ‘Golden Delicious’ cultured for 10 weeks in RITA^®^ bioreactors. (**a**) Shoot rooted spontaneously in the multiplication medium. (**b**,**c**) Aspect of leaves cultured with (**b**) or without activated charcoal. Bars = 1 cm.

**Figure 5 plants-14-02740-f005:**
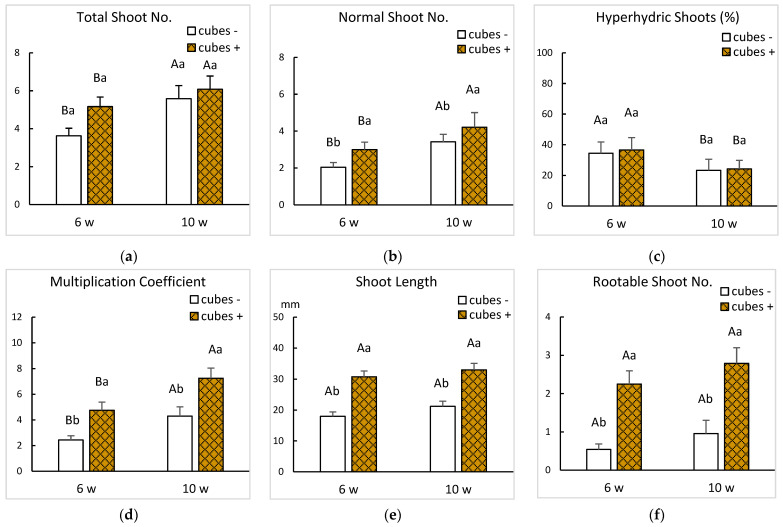
Effect of subculture duration on the growth parameters of Royal Gala shoots cultured in bioreactors with or without cubes. Shoots were immersed 6 times per day for 60 s, and data were recorded 6 and 10 weeks after starting the experiment. (**a**) Number of total shoots per explant. (**b**) Number of normal shoots per explant. (**c**) Percentage of hyperhydric shoots per explant. (**d**) Multiplication coefficient. (**e**) Length of the longest shoot per explant. (**f**) Number of rootable shoots per explant. Means ± standard error were calculated from 3 replicates, each with 8 shoots per treatment (*n* = 24). Different uppercase letters indicate significant differences regarding the duration of subculture, and different lowercase letters indicate significant differences regarding the use of support (*p* ≤ 0.05).

**Figure 6 plants-14-02740-f006:**
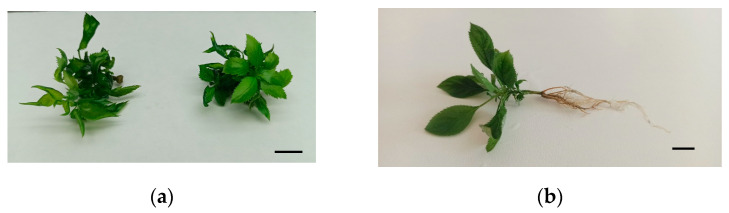
Effect of silver nitrate (SN) and activated charcoal (AC) on ‘Royal Gala’ cultured for 10 weeks in RITA^®^ bioreactors. (**a**) Hyperhydric (left) and normal shoots (right) developed in medium with SN. (**b**). Shoot rooted spontaneously in the multiplication medium with AC.

**Figure 7 plants-14-02740-f007:**
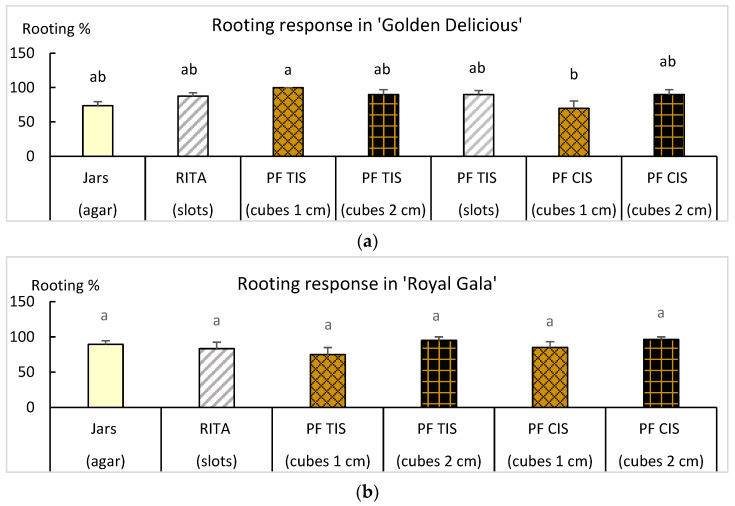
Effect of the culture system on the root formation of ‘Golden Delicious’ (**a**) and ‘Royal Gala’ (**b**). Shoots were treated for 5 weeks with MS ½ macronutrients, 3% sucrose and 4.9 µM indole-3-butyric acid, either in jars with agar or in bioreactors with liquid medium. RITA^®^ bioreactors were operated by TIS (6 immersions per day) and Plantform™ (PF) either by TIS (6 immersions plus 9 aerations) or CIS (15 aerations). Duration of immersions and aerations were 60 s. Shoots were placed between plastic slots (Figure 2) or 1 cm^3^ rockwool cubes or inserted in 2 cm^3^ rockwool cubes. Means ± standard error were calculated from 3 replicates, each with 7 shoots per treatment (*n* = 21). Different letters indicate significant differences (*p* ≤ 0.05).

**Figure 8 plants-14-02740-f008:**
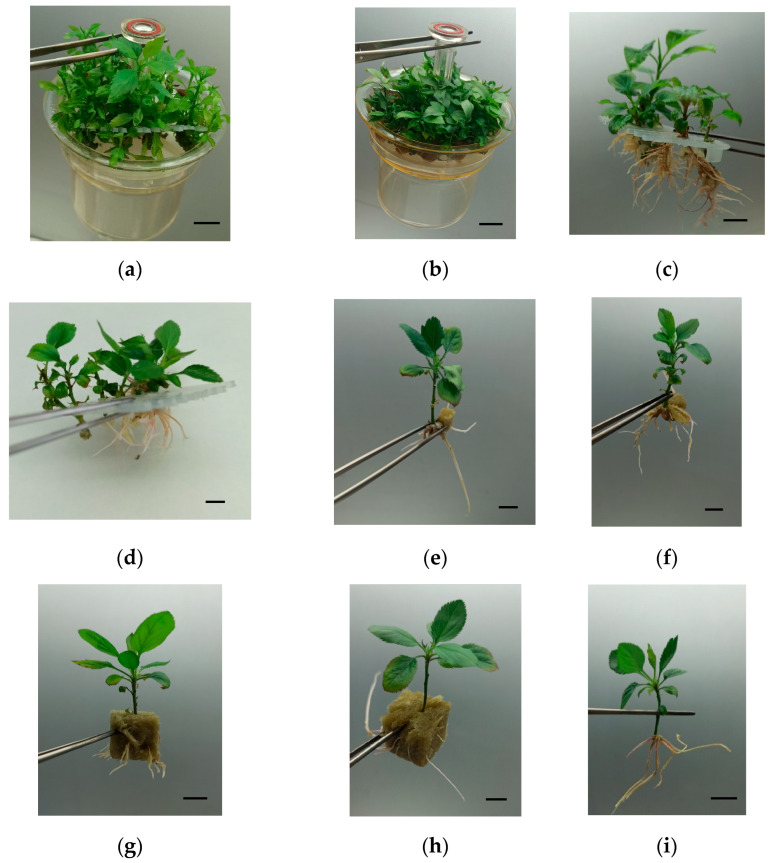
(**a**) Shoots of ‘Golden Delicious’ (GD) used for rooting experiments after 6 weeks in multiplication medium. (**b**) Shoots of ‘Royal Gala’ (RG) used for rooting experiments after 6 weeks in multiplication medium. (**c**–**i**) Rooted shoots 6 weeks after root induction. (**c**,**d**) Shoots of GD (**c**) and RG (**d**) rooted in RITA^®^ with slots. (**e**–**g**) Shoots of GD rooted in Plantform™ by TIS (**e**) and CIS (**f**) with 1 cm^3^ cubes and by CIS inserted in 2 cm^3^ cubes (**g**). (**h**,**i**) Shoots of RG rooted by TIS with 1 cm^3^ (**h**) and 2 cm^3^ cubes (**i**). (**j**,**k**) Rooted shoots of GD (**j**) and RG (**k**) 2 months after transfer to pots. Bars = 1 cm.

**Table 1 plants-14-02740-t001:** Effect of support material on the percentage of new shoots showing hyperhydricity (HH %) and the multiplication coefficient (MC) of apical shoots of ‘Golden Delicious’ and ‘Royal Gala’ cultured in RITA^®^ with 6 immersions of 90 s per day.

Cultivar	[BA]	Support	HH %	MC ^1^
Golden Delicious	4.4 µM	none	26.3	3.7 ± 0.8 b
		slots	11.0	6.6 ± 1.4 a
		cubes	9.4	6.1 ± 0.9 a
Royal Gala	3.1 µM	none	100.0	0.0
		slots	100.0	0.0
		cubes	100.0	0.0

^1^ Means ± standard errors were calculated from 3 replicates, each with 4 shoots per treatment. For MC in ‘Golden Delicious’, different letters indicate significant differences regarding the support (*p* ≤ 0.05). HH % in both cultivars and MC in ‘Royal Gala’ were not statistically analyzed.

**Table 2 plants-14-02740-t002:** Effect of medium, cytokinin concentration and number of immersions per day (duration 60 s) on the percentage of new shoots showing hyperhydricity (HH %) and the multiplication coefficient (MC) of ‘Royal Gala’ cultured in RITA^®^. Each treatment consisted of 8 apical shoots (15–20 mm) that were placed between 1 cm^3^ rockwool cubes. Data were recorded after 6 weeks of culture.

Medium	[BA]	Immersions	HH (%)	MC
MS	3.10 µM	6	100 c	0.00 c
MS	3.10 µM	3	100 c	0.00 c
MS	3.10 µM	2	90 c	0.50 bc
MS	1.55 µM	6	97 c	0.13 c
MS	1.55 µM	3	91 c	0.37 bc
MS	1.55 µM	2	86 c	0.65 b
MS ½ N	3.10 µM	6	92 c	0.37 bc
MS ½ N	3.10 µM	3	75 bc	1.12 b
MS ½ N	3.10 µM	2	67 b	1.13 b
MS ½ N	1.55 µM	6	40 a	3.50 a
MS ½ N	1.55 µM	3	26 a	2.50 a
MS ½ N	1.55 µM	2	20 a	2.00 a

Means with the same letter are not significantly different according to the Kruskal–Wallis test (*p* ≤ 0.05). Percentage data were subjected to arcsine transformation prior to analysis.

**Table 3 plants-14-02740-t003:** Effect of support (no support/cubes) and treatment with 30 µM silver nitrate or 1 g/L activated charcoal on the growth of ‘Golden Delicious’ cultured with BA 4.4 µM in RITA^®^ bioreactors with 6 immersions per day (60 s) for 10 weeks.

Variable	Control		Silver Nitrate		Activated Charcoal	
	No Support	Cubes	No Support	Cubes	No Support	Cubes
Total Shoot No.	5.5 ± 0.49 Aa ^1,2^	5.9 ± 0.52 Aa	3.1 ± 0.42 Ba	3.6 ± 0.56 Ba	1.0 ± 0.04 Ca	1.0 ± 0.00 Ca
Normal Shoot No.	2.7 ± 0.55 Ab	4.8 ± 0.37 Aa	2.9 ± 0.44 Ba	2.9 ± 0.51 Ba	1.0 ± 0.04 Ca	1.0 ± 0.00 Ca
HH Shoots (%)	47 ± 9.1 Aa	13 ± 4.0 Ab	7 ± 4.6 Bb	14 ± 6.6 Ba	0 ± 0.0 Bb	0 ± 0.0 Bb
Shoot Length (mm)	32.3 ± 1.76 Aa	37.6 ± 1.73 Aa	27.3 ± 3.04 Aa	35.7 ± 3.11 Aa	23.8 ± 2.97 Ba	21.0 ± 0.83 Ba
Multiplication Coefficient	4.1 ± 0.78 Ab	8.8 ± 0.75 Aa	4.4 ± 0.78 Ba	4.9 ± 0.96 Ba	1.6 ± 0.16 Ca	1.7 ± 0.09 Ca
Leaf Length (mm)	8.2 ± 0.77 Ca	9.7 ± 0.52 Ca	15.5 ± 1.56 Ba	15.4 ± 1.49 Ba	23.9 ± 1.45 Aa	20.4 ± 1.01 Aa
Leaf Width (mm)	4.6 ± 0.51 Ca	5.3 ± 0.41 Ca	9.1 ± 0.88 Ba	8.7 ± 1.08 Ba	13.4 ± 0.65 Aa	12.3 ± 0.54 Aa
Rootable Shoot No.	1.1 ± 0.24 Ab	3.2 ± 0.26 Aa	1.3 ± 0.33 Ab	2.1 ± 0.37 Aa	0.5 ± 0.10 Ba	0.9 ± 0.06 Ba

^1^ Means ± standard errors were calculated from 3 replicates, each with 8 shoots per treatment (*n* = 24). ^2^ Different uppercase letters indicate significant differences regarding the treatment, and different lowercase letters indicate significant differences regarding the support (*p* ≤ 0.05). When significant interaction between factors occurred, a Bonferroni adjustment was made to detect simple main effects between means.

**Table 4 plants-14-02740-t004:** Effect of support (none/cubes) and treatment with 30 µM silver nitrate (SN) and 1 g/L activated charcoal (AC) alone or combined on the growth of Royal Gala cultured with 1.55 µM in RITA^®^ bioreactors with 6 immersions per day (60 s) for 10 weeks.

Variable ^1^	Control		Silver Nitrate (SN)		Activated Charcoal (AC)		SN + AC	
	No Support	Cubes	No Support	Cubes	No Support	Cubes	No Support	Cubes
TS No.	5.6 ± 0.69 Aa ^2,3^	6.1 ± 0.69 Aa	4.4 ± 0.40 Ba	3.9 ± 0.47 Ba	1.3 ± 0.11 Ca	1.3 ± 0.14 Ca	1.5 ± 0.15 Ca	1.3 ± 0.09 Ca
NS No.	3.4 ± 0.41 Ab	4.2 ± 0.49 Aa	1.5 ± 0.35 Bb	2.3 ± 0.36 Ba	0.8 ± 0.12 Cb	1.3 ± 0.15 Ca	0.9 ± 0.14 Cb	1.0 ± 0.13 Ca
HH S (%)	23 ± 7.2 Ba	24 ± 5.6 Ba	62 ± 8.5 Aa	36 ± 8.1 Ab	33 ± 9.8 Ba	4 ± 4.2 Bb	33 ± 9.1 Ba	17 ± 8.1 Bb
SL (mm)	21.2 ± 1.68 Ab	33.0 ± 2.10 Aa	30.3 ± 2.73 Aa	30.7 ± 2.92 Aa	21.0 ± 1.70 Ba	22.9 ± 1.00 Ba	13.4 ± 0.64 Bb	19.3 ± 0.78 Ba
MC	4.3 ± 0.72 Ab	7.3 ± 0.80 Aa	1.96 ± 0.48 Bb	3.2 ± 0.45 Ba	0.8 ± 0.16 Cb	1.8 ± 0.23 Ca	0.8 ± 0.14 Cb	1.2 ± 0.15 Ca
LL (mm)	7.2 ± 0.48 Cb	8.1 ± 0.38 Ca	10.6 ± 1.94 Bb	11.2 ± 1.21 Ba	18.3 ± 1.52 Ab	21.5 ± 1.12 Aa	16.3 ± 1.21 Ab	22.3 ± 1.84 Aa
LW (mm)	3.6 ± 0.27 Ba	4.3 ± 0.18 Ba	4.2 ± 0.58 Ba	4.8 ± 0.66 Ba	9.4 ± 0.64 Aa	9.9 ± 0.53 Aa	8.2 ± 0.63 Aa	8.5 ± 0.81 Aa
RS No.	1.0 ± 0.35 Ab	2.8 ± 0.40 Aa	0.6 ± 0.21 Bb	1.0 ± 0.18 Ba	0.4 ± 0.10 Bb	0.9 ± 0.06 Ba	0.1 ± 0.06 Bb	0.7 ± 0.10 Ba
SRS No.	0.0 ± 0.0	0.0 ± 0.0	0.04 ± 0.04 Ba	0.04 ± 0.04 Ba	0.58 ± 0.10 Ab	0.92 ± 0.06 Aa	0.65 ± 0.10 Ab	0.74 ± 0.09 Aa

^1^ Variables abbreviations stand for the following: number of total shoots per explant (TS No.); number of normal shoots per explant (NS No.); percentage of hyperhydric shoots per explant (HH S %); length of the longest shoot per explant (SL); multiplication coefficient (MC); length of the largest leaf per explant (LL); width of the largest leaf per explant (LW); number of rootable shoots per explant (RS No.); spontaneously rooted shoot per explant (SRS No.). ^2^ Means ± standard errors were calculated from 3 replicates, each with 8 shoots per treatment (*n* = 24). ^3^ For each variable, different uppercase letters indicate significant differences regarding the treatment, and different lowercase letters indicate significant differences regarding the support (*p* ≤ 0.05). When significant interaction between factors occurred, a Bonferroni adjustment was made to detect simple main effects between means. For SRS No., the two-way ANOVA was performed only with the treatments that produced roots spontaneously during the multiplication step.

## Data Availability

The datasets presented in this article are not readily available because they are part of an ongoing study. Requests to access the datasets should be directed to nieves@mbg.csic.es.

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
