# Peer review of "Micropropagation of Apple Cultivars ‘Golden Delicious’ and ‘Royal Gala’ in Bioreactors"

_plants, 2025, doi:10.3390/plants14172740_

Round 1

Reviewer 1 Report

Comments and Suggestions for Authors

The paper presented by doctor Simón Miranda and colleagues entitled “Micropropagation of apple cultivars ‘Golden Delicious’ and 2 ‘Royal Gala’ in bioreactors” demonstrated the feasibility demonstrated the feasibility of micropropagation using a liquid medium in RITA® bioreactors for the Golden Delicious 625 and Royal Gala cultivars. Authors test different conditions and manage to propagate two varieties, with different success rates. Paper is simple and well written. I would simply suggest simplifying some of the figures to make them more immediate and understandable. 
-Figure 3: use the wording “2.2 vs 4.4” instead of “BA 2.2 μM BA 4.4 μM”, since the comparison is always the same for each parameter considered
-Tables 3 and 4: widen the columns, as in some cases it is difficult to follow the comparisons 

Author Response

Dear Reviewer, thank you for your comments. 

Please sese the attachment with our answers.

Reviewer 2 Report

Comments and Suggestions for Authors

The manuscript presents the use of temporary immersion bioreactors for the micropropagation of the apple cultivars ‘Golden Delicious’ and ‘Royal Gala’. The study evaluated various parameters within the temporary immersion bioreactor system to develop an optimized procedure, which was shown to be genotype-dependent. I have suggested some points for incorporation into the document to improve clarity, consistency, and to facilitate protocol implementation. I have also suggested some sentence rewrites and strongly recommend a review of the punctuation and grammar to enhance the flow of the text. Please find the review report below:

Line 17: Add a comma after “phase”

Line 20: better for xxx – please add the variable

Line 21: Add a comma after “Golden Delicious”

Line 21: were obtained either with

Line 22: for how long?

Line 24: These results were superior to those for ‘Royal Gala’, which

Line 25: rockwool cubes

Line 28: consider adding - immersion, demonstrating the potential of this technology to enhance large-scale plant production for the apple industry.

Line 34: Use the singular form for all fruit species

Line 43: add a reference to support this sentence

Line 44: The high degree of heterozygosity in the genomes of

Lines 48-49: nurseries to produce apple plants is grafting scions onto rootstocks

Line 50: add a reference to support this sentence

Line 53: Consider adding this significant benefit of tissue culture “Additionally, tissue culture protocols play a critical role in the conservation of genetic resources and the production of healthy, disease-free plant material in pome fruit crops.” Suggested references: https://doi.org/10.1094/PHYTO-07-23-0232-KC https://doi.org/10.17660/ActaHortic.2025.1421.16

Line 62: status of xxx

Line 113: Add how old were the cultures

Line 117: Royal Gala’, previously cultured in jars, were placed either directly

Line 137: add the immersion duration and how old were the cultures

Lines 148-150: consider rewriting to “To reduce HH and enable multiplication in this cultivar, we conducted an experiment using rockwool cubes and shortening the immersion duration to 60 s.”

Line 151: µM), and

Line 154: add how old were the cultures

Table 2: how about the statistics?

Line 170: could be mitigated by using

Lines 171-172: consider rewriting to “We then examined the combined effects of reducing cytokinin supplementation and extending the subculture duration from 6 to 10 weeks, together with the use of physical supports.”

Line 184: Figure 3e – add the unit, and adjust the spacing of ‘60’

Line 186: Sometimes it is written as 1 min and in other places as 60 s - they mean the same thing, but use only one format consistently throughout the document.

Line 195:  note that the subtitle is written in a different order.

Lines 196-198: formatting issue

Line 221: add the immersion duration

Tables 3 needs to be formatted so that the treatment information appears on the same line. Just increase the margin and adjust to the text.

Figures 4, 6, and 8 – add scale to the photos

Table 4 – same comment as table 3

Figure 7 – describe PF

Line 377: add the immersion duration

Figure 8 – add how old were the cultures

Line 425: 8.9 and 7.3, and rooting

Line 445: [37], which may prove useful for future management of this condition.

Lines 447-448: Consider rewriting to “Often, the degree of susceptibility is genotype-dependent, as has been reported for pear [39].”

Line 456: kiwifruit

Lines 483-484: In both cultivars, it is desirable to reduce hyperhydricity while maintaining a high multiplication rate and avoiding the use of support elements such as rockwool cubes

Line 489 ammonium [reference]

Line 524: Add a final sentence highlighting the impact and application of this technology.

Lines 527-531: long sentence – please split it into two.

Line 528: in the field?

Line 532: add size of the explants

Line 535: please describe the media and other components present and pH

Line 541: please briefly describe it

Line 545: data – describe what data

Line 550: is it “4.2.1”? in case of positive, revise to the other variables

Line 552: how old? Add this to the text

Line 554: previously autoclaved?? rockwool cubes

Line 570: described by xxxx [15]

Line 587: with leaves? Add this information

Line 625: In this study,

Comments on the Quality of English Language

I have suggested some sentence rewrites and strongly recommend a review of the punctuation and grammar to enhance the flow of the text.

Author Response

Dear Reviewer, than you for your helpful review of our manuscript. Please find the answers in the attachment.

Reviewer 3 Report

Comments and Suggestions for Authors

The manuscript plants-3823068 entitled “Micropropagation of apple cultivars ‘Golden Delicious’ and ‘Royal Gala’ in bioreactors” presents a study on the micropropagation of different apple cultivars using liquid medium in TIS. The experimental design is appropriate and well-structured. Below, please find the comments for your perusal.

In ABSTRACT section,

-Please revise in which stage does the RITA is used and the PLANTFORM because it is confusing.

In INTRODUCTION section,

-Update some references (2024 and 2025).

-The manuscript should more clearly distinguish between the use of RITA and PLANTFORM bioreactors.

-In line 84 and 98 there are some ideas that should be deleted or rewritten.

    Recommendation: The introduction should explicitly define in which stages of micropropagation does the RITA has a better use and in the same way the PLANTFORM bioreactor.

In MATERIALS AND METHODS section,

-Please revise the country and city of reactive and products.

In RESULTS section,

-Please improve some figures. See attached file.

In DISCUSSION section,

-Please explain the role of silver nitrate and activated charcoal.

In REFERENCES section,

-Please revise some errors and change, scientific names should be written in italics.

-Please revise Instruction for Authors.

Author Response

Dear Reviewer, thanks for your comments and suggestions. Please find our answers in the attachment.

Round 2

Reviewer 2 Report

Comments and Suggestions for Authors

The authors did a great job reviewing the document. All the questions and comments have been addressed.

Reviewer 3 Report

Comments and Suggestions for Authors

The authors have addressed all the comments, and the manuscript is now ready for publication.